# Factors associated with stroke among adult patients with hypertension in Ayder Comprehensive Specialized Hospital, Tigray, Ethiopia, 2018: A case-control study

**Haftea Hagos Mekonen**[1]*, **Mulugeta Molla Birhanu**[2], **Tilahun Belete Mossie**[3], **Hagos Tsegabrhan Gebreslassie**[4]

**1** Department of Nursing, College of Medicine and Health Science, Adigrat University, Tigray, Ethiopia,
**2** Department of Nursing, St. Paul's Hospital Millennium Medical College, Addis Ababa, Ethiopia,
**3** Department of Nursing, College of Health Science, Bahirdar University, Tigray, Ethiopia, **4** Department of Nursing, College of Health Science, Mekelle University, Tigray, Ethiopia

☯ These authors contributed equally to this work.
* hafteahagos2@gmail.com

## Abstract

### Background

Globally, the burden of stroke is increasing at an alarming rate. Factors associated with stroke among hypertensive patients are not consistent across different studies and there are limited studies particularly to hypertensive stroke in the particular setting. This study aimed to assess factors associated with stroke among patients with hypertension in Ayder Comprehensive Specialized Hospital, Mekelle, Tigray, Ethiopia, in 2018.

### Methods

Hospital-based case-control study was conducted from February to April 2018. Cases were adult hypertensive patients with stroke and controls were adult hypertensive patients without a stroke. Cases and controls were identified from the patient's card review. Using a systematic random sampling technique 89 cases and 356 controls were included in this study. Record review, physical measurement, and interview techniques were used to collect data. Data was entered and analyzed by using SPSS version 23. Variables with a p-value of less than 0.25 in the bivariate logistic regression were selected for multivariable logistic regression. The adjusted odds ratio and 95% confidence interval were used to determine the association. P-value <0.05 was used to declare statistical significance.

### Results

The mean age of cases and controls were 56.3 years (SD±13.53) and 51.9 years (SD ±12.67) respectively. Lost to follow-up (AOR = 2.474, 95%CI: 1.368–4.929), alcohol drinking (AOR = 2.440, 95%CI: 1.291–4.613), use of excessive salt in diet (AOR = 3.249, 95%CI: (1.544–6.837), medication non-adherence (AOR = 3.967, 95%CI: 2.256–6.973), uncontrolled systolic blood pressure, (AOR = 3.196, 95%CI: 1.60–6.382), uncontrolled diastolic

**Data Availability Statement:** The authors confirm that all data underlying the findings are fully

available without restriction. All relevant data are within the Supporting Information files.

**Funding:** The crossponding author gets fund from Mekelle University (from internal organization) but the funder had no role in study design, data collection and analysis, decision to publish, or preparation of the manuscript.

**Competing interests:** The authors have declared that no competing interests exist.

**Abbreviations:** ACSH, Ayder Comprehensive Specialized Hospital; AOR, Adjusted Odds Ratio; BMI, Body Mass Index; CI, Confidence Interval; CT, Computed Topography scan; DM, Diabetes Mellitus; ECSA, Ethiopian Central Statistical Agency; EDHS, Ethiopian Demographic Health Survey; MRI, Magnetic Resonance Imaging; NGO, Non-Governmental Organizations; OPD, Out Patient Department; OR, Odds Ratio; SPSS, Statistical Package for the Social Science; SSA, Sub Saharan Africa; WHO, World Health Organization.

blood pressure (AOR = 2.204, 95%CI: 1.130–4.297) and high cholesterol level (AOR = 2.413, 95%CI: 1.319–4.414) were found to be significant factors.

## Conclusion

Lost to follow-up, alcohol drinking, uses of excessive salt in diet, medication non-adherence, and uncontrolled systolic and diastolic blood pressure were associated with stroke. Health education on lifestyle practices and hypertension-related complications in each follow-up visit is very essential for improving the primary stroke prevention.

## Introduction

Stroke is a chronic non-communicable disease (NCD) that causes a sudden global focal neurological deficit resulting from infraction or spontaneous hemorrhage in the brain [1]. Stroke leads to multiple social and cognitive impacts like communication difficulty, memory loss, walking difficulty, depression and paralysis [2]. The incidence of a stroke in the past four decades (1970–2010) was increased by 100% in low and middle-income countries, but it was decreased by 42% in developed countries [3]. In 2013, there were 6.5 million stroke deaths, 113 million disability-adjusted life years due to stroke. Developing countries account for 75.2% of all stroke mortality and 81.0% of all stroke-related disability-adjusted life years [4].

In addition to the health consequence cardiovascular disease and stroke has a major impact on economic development. World Economic Forum and world health organization (WHO) forecasts above 7 trillion American dollars will be lost over the period 2011–2025 in low and middle-income countries (LMIC) [5].

The American heart association plans to reduce disease and deaths from stroke by 20 percent in 2020 by focusing on seven key health factors and behaviors that decrease the risk of stroke, those are not-smoking, physical activity, healthy diet, body weight, and control of cholesterol, blood pressure, and blood sugar [6].

Different studies in different parts of the world stated the factors associated with stroke were age, sex, and smoking, low physical exercise, obesity, alcohol, anti-hypertensive medication non-adherence, uncontrolled blood pressure, being diabetic, and cholesterol level [5, 7–12]. But the above factors are different across the studies.

In 2015/16-2019/20 the Ethiopian health sector development program projects to decrease by 12.5% premature mortality from non-communicable disease (NCDs) [13]. In Ethiopia currently, stroke is one of the greatest public health problems, accounts for 7% of total deaths [14]. Study in Mekelle, Ethiopia showed that the stroke was the third most common cause of medical intensive care unit admission (15.2%) and the first cause of death, which accounts for 17% of all deaths in the medical intensive care unit [15]. Similarly, hypertension is responsible for 66.2% of all stroke admission and 38% of all stroke were on anti-hypertensive treatment [16].

In Ethiopia, although admission of stroke patients to the hospitals due to hypertension is increased from time to time, there are limited findings that aim to explore those factors.

Therefore, this study aims to assess the factors associated with stroke among hypertensive patients at Ayder Comprehensive Specialized Hospital, Mekelle, Ethiopia, 2018.

## Materials and methods

### Study setting

Ayder Comprehensive Specialized Hospital (ACSH) is found in Mekelle, Tigray regional state, Ethiopia. Mekelle is found at 783 Km north of Addis Ababa. Ayder Comprehensive Specialized Hospital is the biggest hospital in the region and begins its referral as well as non-referral services in 2008. ACSH serves service for around 9 million populations in its catchment areas of the Tigray, Afar, and parts of the Amhara regional states in Ethiopia. Ayder Comprehensive Specialized Hospital has a capacity of about 500 beds in four major departments and other specialty units. The patient flow of ACSH is above170, 000 per year. The hospital provides hypertensive service in the cardiac unit and stroke service in a neurology unit. Medical ward, medical intensive care unit, cardiac unit and diabetic unit were the study units [17].

### Study design and period

A case-control study design was conducted from February to April 2018.

### Population and sampling

Cases were all sampled adult hypertensive patients with stroke diagnosed by the neurologist (consultant internist) or confirmed by brain imaging (CT-scan) or MRI and Controls were all sampled adult hypertensive patients without clinical evidence of stroke and without a history of stroke available in ACSH during the data collection period. Cases with less than three follow-ups for hypertension treatment before first stroke occurrence and controls with less than three follow-ups for hypertension treatment were excluded. Pregnant mothers were also excluded from both cases and controls. The sample size was calculated using Epi Info version 7 statistical software using the following assumptions: a proportion of 10.7% and 24.1% of greater alcohol consumption was considered for controls and cases respectively [8] at 95% CI, 80% power. The case to control ratio was 1:4. Using those information 81 cases and 324 controls were selected but after adding a 10% non-response rate the total sample becomes 445 of which 89 were cases and 356 were controls. Finally eligible cases and controls recruited using a systematic sampling technique.

### Data collection procedure

Record review was used to identify cases and controls. Information on socio-demographic data and behavioral risk factors for stroke were obtained from the patient or close relative (for unconscious participant) by interview.

Medical history like clinical duration of hypertension, type of stroke, the presence of stroke, and complication other than stroke was taken from the patient record. Height, weight, blood pressure, total cholesterol level and fasting blood sugar were taken during data collection.

During data collection, data collector measures weight, height, blood pressure, as follows. Weight was measured in light closing and without shoes by calibrated UNICEF Seca digital weighing scale. Stadiometer in centimeter in erect position at a precision of 0.1cm without shoes was used to measure height. Mercury sphygmomanometer was used to measure blood pressure average of two measurements 5 minutes apart was recorded for those who we take BP during the data collection.

### Assessment and definition of variables

**Outcome variable:** Stroke

**Independent variables.**

**Scio-demographic included**: age, sex, marital status, occupation, residency and educational status.

**Behavioral factors included:** Physical exercise, Smoking, alcohol, frequency of follow-up, excessive salt in diet, fatty food use, loss to follow-up, medication adherence

physical exercise physically active- if patients make regular physical activities 30 minutes and above, 5 days and above per week physically inactive- if patient is made physical exercise less than 30 minutes per week or less than 5 days per week [18]. Medication adherence was assessed using Morisky medication adherence score to anti-hypertensive medications having eight questions each with yes = 0 and No = l, adherent if they score 7–8 and non-adherent if they score < = 6 [19].

Alcohol drinker- a person who drinks 10.5 units of alcohol and above per week [18].

**Physical measurements and clinical factors:** Fasting blood glucose (FBG), cholesterol level, blood pressure control, body mass index (BMI) and comorbidities. Normal FBG <126 mg/dl, raised FBG > = 126 mg/dl [8].

Cholesterol level: normal if less than 200 and high cholesterol level 200 and above, BMI: underweight (less than 18.5), normal (18.5–24.9), overweight (25–29.9) and obese (30 and above). Systolic blood pressure: controlled (<140) and uncontrolled (> = 140), diastolic blood pressure: controlled (<90) and uncontrolled (> = 90) [20].

## Data analysis and management

Data were cleaned, coded, entered and analyzed using SPSS version 23. Summary statistics: frequencies and tables were used to present for categorical variables and mean for continuous variables in both cases and controls.

First bivariate logistic regression was done to assess the association between each independent variable and the dependent variable. Variable with a P-value < 0.25 significance level in bivariate logistic regression was taken to multivariable logistic regression. Finally, multivariable logistic regression was used to assess the association between independent variables with the dependent variable and to control confounding variables. Adjusted odds ratio and P-value <0.05 and with 95% CI were used to declare statistical significance.

## Ethics approval and consent to participate

Ethical clearance was obtained from Mekelle University, College of health science institutional review board (ERC 1295/2018). Official permission was obtained from ACSH chief executive director and study participants were informed about the purpose of the study. The information was collected after obtaining written informed consent from the participant (relatives for patients who were critically ill). The respondents were informed as they have the right to refuse or discontinue participation at any time if unwanted. The information was recorded anonymously and confidentiality and beneficence were assured throughout the study period.

## Results

### Socio-demographic characteristics of the respondents

All 445 selected participants (89 cases and 356 controls) have participated in the study and the response rate was 100%. The mean age of cases was 56.3years (SD±13.53) and 51.9 years (SD ±12.67) for controls.

The majority of subjects 64% in cases and 66% in controls were married. Thirty (33.3%) of the cases and 117(32.9%) of controls were self-employed (Table 1).

**Table 1. Socio demographic characteristics of the study participants.**

| Variables | Categories | Control | | Cases | |
|---|---|---|---|---|---|
| | | Frequency | Percentage | Frequency | Percentage |
| Sex | Male | 170 | 47.8 | 46 | 51.7 |
| | Female | 186 | 52.2 | 43 | 48.3 |
| Age(in years) | <45 | 116 | 32.6 | 23 | 25.8 |
| | 45–65 | 185 | 52 | 39 | 43.8 |
| | >65 | 55 | 15.4 | 27 | 30.3 |
| Religion | Orthodox Christian | 278 | 78.1 | 69 | 77.5 |
| | Muslim | 61 | 17.1 | 17 | 19.1 |
| | Other | 17 | 4.8 | 3 | 3.4 |
| Ethnicity | Tigray | 316 | 88.8 | 81 | 91 |
| | Amhara | 22 | 6.2 | 4 | 4.5 |
| | Afar | 15 | 4.2 | 4 | 4.5 |
| | Other | 3 | 0.8 | - | - |
| Marital status | Married | 235 | 66 | 57 | 64 |
| | Single | 64 | 18 | 16 | 18 |
| | Divorce | 25 | 7 | 6 | 6.7 |
| | Widowed | 32 | 9 | 10 | 11.2 |
| Educational status | No formal education | 138 | 38.8 | 31 | 34.8 |
| | Primary school | 65 | 18.3 | 25 | 29.2 |
| | Secondary school | 42 | 11.8 | 9 | 10.1 |
| | Diploma | 29 | 8.1 | 6 | 6.7 |
| | University and above | 82 | 23 | 17 | 19.1 |
| Occupation | Farmer | 73 | 20.5 | 15 | 16.9 |
| | Household | 27 | 7.6 | 5 | 5.6 |
| | Governmental employee | 100 | 28.1 | 20 | 22.5 |
| | Non-Governmental employee | 32 | 9 | 15 | 16.9 |
| | Self-employee | 117 | 32.9 | 30 | 33.7 |
| | Other | 7 | 2 | 4 | 4.5 |
| Residency | Rural | 85 | 23.9 | 21 | 24.7 |
| | Urban | 271 | 76.1 | 67 | 75.3 |

## Behavioral factors of the respondents

Out of 89 cases and 356 controls, 32(36%) of cases and 49(13.8%) controls were current alcohol drinkers. Twenty-two (24.7%) cases and 28(7.9%) of controls did not reduce salt in their diet. In this study, 66.3% of the cases and 29.8% of controls were non-adherent to medication. Sixty-three (70.8%) and 245(68.8%) controls were not on regular exercise. Thirty two (36%) of cases and 46 (12.9%) of controls had lost to follow up. Fifty-two (58.4%) of cases and 215 (60.4%) of controls had every two months follow-up (Table 2).

## Physical measurements and clinical characteristics of the respondents

Among the study participants, 13 (14.6%) of cases and 60 (16.9%) of controls were overweight. The clinical characteristics of patients showed that 5 (5.6%%) of cases and 25 (7%) of controls of the participants had a family history of stroke (Table 3).

The mean of the clinical duration of hypertension was 4.65 ± 3.3 years for cases and 3.94± 3.18 years for controls. The mean of total cholesterol was 198±34 among cases and 182±27 for controls. The mean systolic blood pressure was 150±14 in the case and 145±17 in the controls. The mean diastolic blood pressure was 92±7 in the case and 90±9 in the controls.

**Table 2. Behavioral factors results of the participants.**

| Variables | Categories | Control | | Cases | |
|---|---|---|---|---|---|
| | | Frequency | Percentage | Frequency | Percentage |
| Have you ever smoke cigarette | Yes | 10 | 2.8 | 2 | 2.2 |
| | No | 346 | 97.2 | 87 | 97.8 |
| Smoking after you diagnosed hypertension | Yes | 3 | 0.8 | 0 | 0 |
| | No | 353 | 99.2 | 89 | 100 |
| Regular physical exercise | Yes | 111 | 31.2 | 26 | 29.2 |
| | No | 245 | 68.8 | 63 | 70.8 |
| Medication non-adherence | Yes | 106 | 29.8 | 59 | 66.3 |
| | No | 250 | 70.2 | 30 | 33.7 |
| Have ever drink alcohol | Yes | 255 | 71.6 | 67 | 75.3 |
| | No | 101 | 28.4 | 22 | 24.7 |
| Current alcohol drink | Yes | 49 | 13.8 | 32 | 36 |
| | No | 307 | 86.2 | 57 | 64 |
| Do reduce salt in diet | Yes | 328 | 92.1 | 67 | 75.3 |
| | No | 28 | 7.9 | 22 | 24.7 |
| Do you eat fatty diet | Yes | 71 | 19.9 | 27 | 30.3 |
| | No | 285 | 80.1 | 62 | 69.7 |
| Lost to follow-up | Yes | 46 | 12.9 | 32 | 36 |
| | No | 310 | 87.1 | 57 | 64 |
| Frequency of follow-up | 1month | 85 | 23.9 | 21 | 23.6 |
| | 2month | 215 | 60.4 | 52 | 58.4 |
| | 3month | 56 | 15.7 | 16 | 18 |

**Table 3. Clinical and anthropometric measurements of the study participants.**

| Variables | Categories | Controls | | Cases | |
|---|---|---|---|---|---|
| | | Frequency | Percentage | Frequency | Percentage |
| Duration of diagnosis HTN | <4year | 249 | 69.9 | 53 | 59.6 |
| | > = 4year | 107 | 30.1 | 36 | 40.4 |
| Cholesterol level | Normal | 282 | 79.2 | 56 | 62.9 |
| | High level | 74 | 20.8 | 33 | 37.1 |
| Blood glucose level | Normal | 304 | 85.4 | 75 | 84.3 |
| | High level | 52 | 14.6 | 14 | 15.7 |
| BMI | 18.5–24.9 | 288 | 80.9 | 74 | 83.1 |
| | 25–29.9 | 60 | 16.9 | 13 | 14.6 |
| | > = 30 | 8 | 2.2 | 2 | 2.2 |
| Systolic BP | Controlled | 156 | 43.8 | 15 | 16.9 |
| | Uncontrolled | 200 | 56.2 | 74 | 83.1 |
| Diastolic BP | Controlled | 163 | 45.8 | 18 | 20.2 |
| | uncontrolled | 193 | 54.2 | 71 | 79.8 |
| Comorbidities | Yes | 67 | 19.5 | 22 | 24.7 |
| | No | 277 | 80.5 | 67 | 75.3 |
| Family history of stroke | Yes | 25 | 7 | 5 | 5.6 |
| | No | 331 | 93 | 84 | 94.4 |

**Table 4. Bivariate and multivariable logistic regression result of the study.**

| Variables | Category | Cases% | Controls% | COR (95%CI) | AOR(95%CI) |
|---|---|---|---|---|---|
| Age (years) | <45 | 23(25.8%) | 116(32.6% | 1 | 1 |
| | 45–65 | 39(43.8%) | 185(52%) | 1.063(0.604–1.871) | .811(.418–1.573) |
| | >65 | 27(30.3%) | 55(15.4%) | 2.476(1.303–4.705)* | 1.779(.836–3.787) |
| Have you ever lost to follow up | Yes | 32(36%) | 46(12.9%) | 3.783(2.222–6.442)* | 2.59(1.368–4.929)* |
| | No | 57(64%) | 310(87.1%) | 1 | 1 |
| Have you drink alcohol after you diagnosed | Yes | 32(36%) | 49(13.8%) | 3.517(2.075–5.961)* | 2.44(1.291–4.613)* |
| | No | 57(64%) | 307(86.2%) | 1 | 1 |
| Do you reduce salt in diet | Yes | 67(75.3%) | 328(92.1%) | 1 | 1 |
| | No | 22(24.7%) | 28(7.9%) | 3.846(2.075–7.130)* | 3.25(1.544–6.837)* |
| Do you eat fatty foods | Yes | 27(30.3%) | 71(19.9%) | 1.748(1.038–2.944)* | 1.39(.733–2.63) |
| | No | 62(69.7%) | 285(80.1%) | 1 | 1 |
| Medication adherence | No adherent | 59(66.3%) | 106(29.8%) | 4.638(2.828–7.607)* | 3.97(2.256–6.973)* |
| | Adherent | 30(33.7%) | 250(70.2%) | 1 | 1 |
| Cholesterol level | Normal | 56(62.9%) | 282(79.2%) | 1 | 1 |
| | High level | 33(37.1%) | 74(20.8%) | 4.560(2.313–8.989)* | 2.41(1.319–4.414)* |
| Systolic blood pressure | Controlled | 15(16.9%) | 156(43.8%) | 1 | 1 |
| | Uncontrolled | 74(83.1%) | 200(56.2%) | 3.848(2.126–6.964)* | 3.19(1.60–6.382)* |
| Diastolic blood pressure | Controlled | 18(20.2%) | 163(45.8%) | 1 | 1 |
| | Uncontrolled | 71(79.8%) | 193(54.2%) | 3.331(1.907–5.818)* | 2.2(1.130–4.297) |

## Types of stroke and method used to diagnosis

The type of stroke (ischemic or hemorrhagic) and the tool by what they identified could be diagnosed was from the patient's chart. Out of 89 cases, 29 were ischemic stroke cases and 60 were a hemorrhagic stroke. From all stroke 80% were diagnosed by CT scan, 4.5% by MRI and 15.7% clinically.

## Bivariate and multivariable logistic regression for factors associated with stroke among hypertensive patients

The bivariate logistic regression result reveals that age, lost to follow-up, alcohol drinkers after he/she knows their hypertensive status, use of excessive salt in diet, use of fatty diet, medication non-adherence, high cholesterol level, and uncontrolled systolic and diastolic blood pressure were found be significant predictors of stroke (Table 4).

Multivariable logistic regression lost to follow up (AOR = 2.474, 95% CI: 1.368–4.929), alcohol drinkers after he/she know their hypertensive status (AOR = 2.440, 95%CI: 1.291–4.613), use of excessive salt- in diet (AOR = 3.249, 95%CI: 1.544–6.837), medication non-adherence (AOR = 3.967, 95%CI: 2.256–6.973), high cholesterol level (AOR = 2.413, 95%CI: 1.319–4.414), uncontrolled systolic (AOR = 3.196, 95%CI: 1.60–6.382) and diastolic blood pressure (AOR = 2.204, 95%CI: 1.130–4.297) were found be significant predictors of stroke(Table 4).

## Discussion

This case-control study aimed to identify factors associated with stroke among hypertensive patients in Ayder comprehensive specialized hospital, Tigray, north Ethiopia. Lost to follow-up, current alcohol drinking, eating salty foods, medication non- adherence, high cholesterol level, uncontrolled systolic and diastolic blood pressure were significant factors for stroke.

In this study, patients who had a history of loss to follow up were 2.5 times more likely to be at high risk for stroke than their counterparts (AOR = 2.474, 95% CI: 1.368–4.929). This finding is in agreement with a study conducted in Germany [21]. This similarity might be due to missing their routine medications and lifestyle modification counseling leads to uncontrolled hypertension.

Hypertensive patients, who were alcohol drinkers, were 2.44 times more likely to be at high risk for stroke (AOR = 2.440, 95%CI: 1.291–4.613). This finding is in line to a study conducted in Nigeria [8] and in 32 countries (INTERSTROKE) [7]. This similarity might be due to alcohol has a direct impact on raising blood pressure.

Hypertensive patients who did not reduce salt in diet were around 3.2 times more likely to be at high risk for stroke (AOR = 3.249, 95%CI: 1.544–6.837). This is similar with a study done in 32 countries (INTERSTROKE) [7]. It is due to the fact salt has an impact on raising blood pressure on circulation then can cause a stroke.

In this study medication none-adhered were 4 times (AOR = 3.967, 95%CI: 2.256–6.973) more likely to be at high risk for stroke than the medication adherent hypertensive patients. This finding is in line with a study done in Bangladesh, Germany [21, 22].

Patients with high cholesterol levels were 2.4 times (AOR = 2.413, 95%CI: 1.319–4.414) more likely to be at high risk for stroke than patients with low cholesterol levels. This finding is consistent with a study conducted in Nigeria and Tanzania [8, 23]. This might due to cholesterol have a direct impact on block blood circulation and can cause a stroke. But study Erbil reveals that there was no statistically significant difference between the groups with and without stroke with respect to Cholesterol variation [9]. This might be due to sample size, study design difference.

Result of this study showed that uncontrolled systolic blood pressure is 3.2 times more likely at high risk for the development of stroke (AOR = 3.196, 95%CI: 1.60–6.382) and uncontrolled diastolic blood pressure were 2.2 times more likely for the development of stroke (AOR = 2.204, 95%CI: 1.130–4.297) this in lines with a study done in Puget [24]. This might be related to the fact that uncontrolled blood pressure cause hemorrhage in the brain and blockage in the blood vessel.

## Conclusion

Among hypertensive patients, alcohol consumption loss to follow up, excessive salt use in diet, high cholesterol level, uncontrolled systolic and diastolic blood pressure were found to be associated with stroke. Therefore, further intervention and prevention mechanism are essential for improving the primary stroke prevention.

## Limitations of the study

Control subjects were not recruited from the general population, so selection bias of control subjects may also have affected the findings of this study. Two blood pressure readings were taken from the patient's record review hence no information was available on how BP was measured. A case-control study design was used which does not allow for the temporal relationship to be established.

## Supporting information

**S1 Dataset. SPSS data of the questionnaire results.**
(XLSX)

**S1 Appendix. English and Tigrigna version questionnaires of the study.**
(DOCX)

# Acknowledgments

First of all, I would like to thank you for Mekelle University for financial and technical support. I would also like to thank you for data collectors, supervisors and study participants for their great effort in acquiring necessary information.

# Author Contributions

**Conceptualization:** Haftea Hagos Mekonen, Mulugeta Molla Birhanu.

**Data curation:** Haftea Hagos Mekonen.

**Formal analysis:** Haftea Hagos Mekonen, Mulugeta Molla Birhanu, Tilahun Belete Mossie.

**Investigation:** Haftea Hagos Mekonen, Hagos Tsegabrhan Gebreslassie.

**Methodology:** Haftea Hagos Mekonen, Mulugeta Molla Birhanu.

**Supervision:** Haftea Hagos Mekonen, Mulugeta Molla Birhanu, Hagos Tsegabrhan Gebreslassie.

**Writing – original draft:** Haftea Hagos Mekonen.

**Writing – review & editing:** Mulugeta Molla Birhanu, Tilahun Belete Mossie, Hagos Tsegabrhan Gebreslassie.

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
