## [Decision Letter · Decision Letter 0]

12 Dec 2019

PONE-D-19-31261

Factors associated with Stroke among Adult Patients with Hypertension in Ayder Comprehensive Specialized Hospital, Tigray, Ethiopia, 2018 A Case Control Study

PLOS ONE

Dear Dr. Mekonen,

Thank you for submitting your manuscript to PLOS ONE. After careful consideration, we feel that it has merit but does not fully meet PLOS ONE’s publication criteria as it currently stands. Therefore, we invite you to submit a revised version of the manuscript that addresses the points raised during the review process.

We would appreciate receiving your revised manuscript by Jan 26 2020 11:59PM. To enhance the reproducibility of your results, we recommend that if applicable you deposit your laboratory protocols in protocols.io, where a protocol can be assigned its own identifier (DOI) such that it can be cited independently in the future. For instructions see: http://journals.plos.org/plosone/s/submission-guidelines#loc-laboratory-protocols

We look forward to receiving your revised manuscript.

Kind regards,

HASNAIN SEYED EHTESHAM

Academic Editor

PLOS ONE

Journal Requirements:

**When submitting your revision**, **we need you to address these additional requirements:**

**Please ensure that your manuscript meets PLOS ONE's style requirements, including those for file naming**. **The PLOS ONE style templates can be found at http://www.plosone.org/attachments/PLOSOne_formatting_sample_main_body.pdf and http://www.plosone.org/attachments/PLOSOne_formatting_sample_title_authors_affiliations.pdf**In your Methods section please provide a rationale for excluding pregnant women.Thank you for stating the following financial disclosure:

[no funding].

Please provide an amended Funding Statement that declares *all* the funding or sources of support received during this specific study (whether external or internal to your organization) as detailed online in our guide for authors at http://journals.plos.org/plosone/s/submit-now. 

Please state what role the funders took in the study.  If any authors received a salary from any of your funders, please state which authors and which funder. If the funders had no role, please state: "The funders had no role in study design, data collection and analysis, decision to publish, or preparation of the manuscript."

5. Your ethics statement must appear in the Methods section of your manuscript. If your ethics statement is written in any section besides the Methods, please move it to the Methods section and delete it from any other section. Please also ensure that your ethics statement is included in your manuscript, as the ethics section of your online submission will not be published alongside your manuscript.

6.Please include your tables as part of your main manuscript and remove the individual files. Please note that supplementary tables (should remain/ be uploaded) as separate "supporting information" files

8. We note that you have stated that you will provide repository information for your data at acceptance. Should your manuscript be accepted for publication, we will hold it until you provide the relevant accession numbers or DOIs necessary to access your data. If you wish to make changes to your Data Availability statement, please describe these changes in your cover letter and we will update your Data Availability statement to reflect the information you provide.

9.   We suggest you thoroughly copyedit your manuscript for language usage, spelling, and grammar. If you do not know anyone who can help you do this, you may wish to consider employing a professional scientific editing service.  

Additional Editor Comments (if provided):

Major Revision

Reviewers' comments:

Reviewer's Responses to Questions

**Comments to the Author**

1. Is the manuscript technically sound, and do the data support the conclusions?

Reviewer #1: Yes

Reviewer #2: No

2. Has the statistical analysis been performed appropriately and rigorously? 

Reviewer #1: Yes

Reviewer #2: I Don't Know

3. Have the authors made all data underlying the findings in their manuscript fully available?

Reviewer #1: Yes

Reviewer #2: No

4. Is the manuscript presented in an intelligible fashion and written in standard English?

Reviewer #1: Yes

Reviewer #2: No

5. Review Comments to the Author

Reviewer #1: Manuscript #: PONE–D- 31261

Title: Factors associated with stroke among Adult patients with Hypertension in Ayder Comprehensive Specialized Hospital, Tigray, Ethiopia, 2018. A Case Control study.

Review:

1. The article is well written. However, since the authors are working on the factors associated with stroke, a range of the age group taken (Minimum & maximum age in adults) instead of the mean age would be more informative to further identify the incidence of stroke, whether more common in young adults or older adults.

2. The inclusion of some more parameters like Gender, Socioeconomic status, dietary habits (Consumption of more Fatty or non fatty food), profession (Stressful/ non stressful), life style (Sedentary/ Active), & family history will make this study and identification of factors more meaningful.

Reviewer #2: I have gone through the manuscript entitled "Factors associated with Stroke among Adult Patients with Hypertension in Ayder Comprehensive Specialized Hospital, Tigray, Ethiopia, 2018 A Case Control Study". The article in contention is not acceptable in its present form.

Comments:

1. Usage of English language is not up to the mark, there are several spelling mistakes, grammatical and typo error in this manuscript.

2. There is only one reference of 2018 and no reference of 2019 and the referencing is not as per journal format.

3. Annexure I is only available but other supporting tables are not in readable format.

6. PLOS authors have the option to publish the peer review history of their article (what does this mean?). If published, this will include your full peer review and any attached files.

Reviewer #1: No

Reviewer #2: No

---

## [Author Response · Author response to Decision Letter 0]

14 Jan 2020

Response to Reviewers

PONE-D-19-31261

Title: Factors associated with Stroke among Adult Patients with Hypertension in Ayder Comprehensive Specialized Hospital, Tigray, Ethiopia, 2018: A case-control study

PLOS ONE

Point by point response to Reviewers Comments:-

Sr. No. Reviewers' Comments Response to Comments

 Academic Editor comments 

1 Please ensure that your manuscript meets PLOS ONE's style requirements, including those for file naming. The PLOS ONE style templates can be found at http://www.plosone.org/attachments/PLOSOne_formatting_sample_main_body.pdf and http://www.plosone.org/attachments/PLOSOne_formatting_sample_title_authors_affiliations.pdf

We accept the comment and we made correction accordingly. 

2 In your Methods section please provide a rationale for excluding pregnant women. Pregnant women were excluded because: 

1. Hypertensive mothers increases the risk of stroke during pregnancy due to the pregnancy itself increase the chance of stroke because of physiological effect (hormonal change) so, this physiological change can affect the our result.

2. Anthropometric measurement(weight) is difficult in pregnant mother in order to calculate body mass index (BMI) of the participant.

3 Please provide an amended Funding Statement that declares *all* the funding or sources of support received during this specific study (whether external or internal to your organization) as detailed online in our guide for authors at http://journals.plos.org/plosone/s/submit-now. 

 The crossponding author gets fund from Mekelle University (from internal organization) but the funder had no role in study design, data collection and analysis, decision to publish, or preparation of the manuscript."

4 PLOS requires an ORCID iD for the corresponding author in Editorial Manager on papers submitted after December 6th, 2016. Please ensure that you have an ORCID iD and that it is validated in Editorial Manager. I have created ORCID iD and validated in the editorial manager.

5 Your ethics statement must appear in the Methods section of your manuscript. If your ethics statement is written in any section besides the Methods, please move it to the Methods section and delete it from any other section. Please also ensure that your ethics statement is included in your manuscript, as the ethics section of your online submission will not be published alongside your manuscript We corrected it. Ethical statement is now in methods section

6 Please include your tables as part of your main manuscript and remove the individual files. Please note that supplementary tables (should remain/ be uploaded) as separate "supporting information" files We accepted your comment. Now tables are within the main manscript and individual tables are removed. 

7 Please include captions for your Supporting Information files at the end of your manuscript, and update any in-text citations to match accordingly. Please see our Supporting Information guidelines for more information: http://journals.plos.org/plosone/s/supporting-information

We accept the comments and sugestions. We made correction accordingly 

8 We note that you have stated that you will provide repository information for your data at acceptance. Should your manuscript be accepted for publication, we will hold it until you provide the relevant accession numbers or DOIs necessary to access your data. If you wish to make changes to your Data Availability statement, please describe these changes in your cover letter and we will update your Data Availability statement to reflect the information you provide. We changed our idea and the data is now available in "supporting information" files

9 We suggest you thoroughly copyedit your manuscript for language usage, spelling, and grammar. If you do not know anyone who can help you do this, you may wish to consider employing a professional scientific editing service. We accept the comment and we made corrections regarding the grammatical and punctuation errors throughout the whole document.

 Reviewer#1 

1 The article is well written. However, since the authors are working on the factors associated with stroke, a range of the age group taken (Minimum & maximum age in adults) instead of the mean age would be more informative to further identify the incidence of stroke, whether more common in young adults or older adults. The range of the age group we take is based on previous similar published articles. We searched many articles to get the appropriate age categories finally we believe that this age categories was suitable for our study. In the previous studies age greater than 65 years were high risk for stroke and they used similar categories of age. 

2 The inclusion of some more parameters like Gender, Socioeconomic status, dietary habits (Consumption of more Fatty or non fatty food), profession (Stressful/ non stressful), life style (Sedentary/ Active), & family history will make this study and identification of factors more meaningful. We assessed sex (Male, Female), dietery habits (Did you eat diet high in fat such as Fatty meal and animal product)(Yes, No), profession(occupation)(Farmer, House wife, Government employee, Non-Government organization employee, Self-employee, Other), physical exercise was assed using two questions (i.e How many days do you do regular physical exercise in a week __and On average how much time do you spend during those exercise in a typical day_) finally it was diactomized in to physically active and phyisically inactive and family history of stroke(Yes, No). The socio economic status of the participants, actualy average montlty income was in our questioner but almost in all the questioner the value was missing it may be due to most patients have no fixed monthly income especially participants from the rular area finally due to high missing value we didn’t consider for analysis.

 Reviewer#2 

1 Usage of English language is not up to the mark, there are several spelling mistakes, grammatical and typo error in this manuscript. We accept the comments and we made corrections on the grammatical and punctuation errors throughout the whole document.

2 There is only one reference of 2018 and no reference of 2019 and the referencing is not as per journal format. We made uptodated informations in our paper. 

3 Annexure I is only available but other supporting tables are not in readable format. I accepted your comment. Now tables are within the main manscript and individual tables are removed. We made corrections based on the journal format and in areadable form.

---

## [Editor Report · Decision Letter 1]

22 Jan 2020

Factors associated with Stroke among Adult Patients with Hypertension in Ayder Comprehensive Specialized Hospital, Tigray, Ethiopia, 2018 A Case Control Study

PONE-D-19-31261R1

Dear Dr. mekonen,

We are pleased to inform you that your manuscript has been judged scientifically suitable for publication and will be formally accepted for publication once it complies with all outstanding technical requirements.

With kind regards,

HASNAIN SEYED EHTESHAM

Academic Editor

PLOS ONE

Additional Editor Comments (optional):

I have gone through the revised manuscript and also the author's response to the comments of the reviewers. Clarification about age group selection is acceptable. Similarly, inclusion criterion has been explained. Grammatical and language corrections have been made; tables have become more understandable. I recommend publication of this manuscript.
---

## [Editor Report · Acceptance letter]

4 Feb 2020

PONE-D-19-31261R1 

Factors associated with Stroke among Adult Patients with Hypertension in Ayder Comprehensive Specialized Hospital, Tigray, Ethiopia, 2018: A Case-Control Study 

Dear Dr. Mekonen:

I am pleased to inform you that your manuscript has been deemed suitable for publication in PLOS ONE. Congratulations! Your manuscript is now with our production department. 

With kind regards,

on behalf of

Prof HASNAIN SEYED EHTESHAM 

Academic Editor

PLOS ONE